# Dynamics of homeostats: the basis of electrical, chemical, hydraulic, pH and calcium signaling in plants

## Theories

homeostasis; membrane transport; modeling; quantitative biology; thermodynamics; transporter networks.

**Corresponding author:**
Ingo Dreyer;
Email: idreyer@utalca.cl

Leslie Contador-Álvarez, Tamara Rojas-Rocco, and Talía Rodríguez-Gómez these authors contributed equally.

**Associate Editor:**
Richard Morris

Leslie Contador-Álvarez[1], Tamara Rojas-Rocco[1], Talía Rodríguez-Gómez[1], María Eugenia Rubio-Meléndez[2] , Janin Riedelsberger[2] , Erwan Michard[3] and Ingo Dreyer[2]

[1]Programa de Doctorado en Ciencias mención Modelado de Sistemas Químicos y Biológicos, Universidad de Talca, Talca, Chile; [2]Electrical Signaling in Plants (ESP) Laboratory–Center of Bioinformatics, Simulation and Modeling (CBSM), Faculty of Engineering, Universidad de Talca, Talca, Chile; [3]Instituto de Ciencias Biológicas, Universidad de Talca, Talca, Chile

## Abstract

Homeostats are important to control homeostatic conditions. Here, we have analyzed the theoretical basis of their dynamic properties by bringing the K homeostat out of steady state (i) by an electrical stimulus, (ii) by an external imbalance in the $K^+$ or $H^+$ gradient or (iii) by a readjustment of transporter activities. The reactions to such changes can be divided into (i) a short-term response (tens of milliseconds), where the membrane voltage changed along with the concentrations of ions that are not very abundant in the cytosol ($H^+$ and $Ca^{2+}$), and (ii) a long-term response (minutes and longer) caused by the slow changes in $K^+$ concentrations. The mechanistic insights into its dynamics are not limited to the K homeostat but can be generalized, providing a new perspective on electrical, chemical, hydraulic, pH and $Ca^{2+}$ signaling in plants. The results presented here also provide a theoretical background for optogenetic experiments in plants.

## 1. Introduction

Homeostasis is the ability of organisms to maintain a stable internal state by compensating for changes in their environment through a regulated exchange of matter and energy with the exterior. An important example is ion homeostasis in plants (Serrano & Rodriguez-Navarro, 2001). Homeostatic ion balance is established through the coordinated activity of membrane transporters that act together in transporter networks. The networks display the inherent property of defining the transmembrane concentration gradient in a steady state depending on the relative activity of the individual transporters. We previously proposed to call these remarkable transporter network homeostats (Dreyer, 2021). A homeostat comprises differently energized types of transporters that are permeable for a specific ion species, either alone or together with other ions, for example, $H^+$. Unbiased, systemic modeling approaches combining thermodynamics with biophysics and mathematics showed that the homeostatic property of homeostats emerges from the coupling of the transporters' activity through the membrane voltage and the electrochemical gradients of the permeating ions. In a previous report, we presented a hands-on tutorial for modeling a homeostat in (dynamic) equilibrium using the potassium (K) homeostat as an example (Dreyer et al., 2024). We derived analytical solutions and analyzed the K homeostat in computational simulations in its steady state. However, steady-state conditions are only part of the life of a plant cell.

Here, we now analyze the homeostat out of equilibrium and monitor its dynamic behavior. Using again the example of the K homeostat, we present the mathematical background that describes its deflection from a steady state and how the thermodynamic constraints determine its reactions to perturbations. In particular, we analyzed the responses to

(i) An electrical stimulus such as occurs during an action potential, for instance,
(ii) An imbalance in the $K^+$ or $H^+$ gradients caused, for example, by environmental stress,

(iii) A change in transporter activities, for example, caused by posttranslational modifications such as (de)phosphorylation but also during hydraulic or chemical signaling.

The detour via mathematics and computer simulations reveals again far-reaching insights into fundamental physiological properties of homeostats in general (not only the K homeostat) that go beyond the mere establishment of homeostasis. The results strongly suggest that homeostats are deeply involved in electrical, chemical, hydraulic, pH and calcium signaling in plants.

The study of the K homeostat under nonequilibrium conditions could also provide important impetus for agriculture, as $K^+$ plays a variety of important physiological roles (see the article on $K^+$ homeostasis in this collection; Wegner et al., 2025). Most of these functions are indirect effects of $K^+$ due either to its concentration (osmolarity and electrostatic effects) or its fluxes (membrane voltage, indirect effect on pH and mass fluxes of ions). It is not always easy to understand the indirect effects; this is where mathematical modeling of homeostats can help. More fundamentally, the quantification of the activity of homeostats and its change through indirect physiological parameters such as voltage, concentration, pressure and pH requires numerical quantification, which is only possible through mathematical modeling. Understanding how transporter networks in plants respond to adverse conditions is important for developing strategies to improve crop resilience so that they can thrive in a changing environment.

## 2. Methods

The mathematical description of the K homeostat and its analysis in steady state have been presented in detail in Dreyer et al. (2024). In short, the K homeostat (Supplementary Figure S1, Supplementary Table S1) is composed of $K^+$ channels (KC), $H^+/K^+$ symporters (HKs), $H^+/K^+$ antiporters (HKa) and $H^+$-ATPases, embedded in a membrane with membrane capacitance and membrane voltage ($C$, $V$). The membrane separates the internal and external compartments (volumes $Vol_{in}$, $Vol_{out}$) with respective proton and potassium concentrations ($[H^+]_{in}$, $[H^+]_{out}$, $[K^+]_{in}$, $[K^+]_{out}$). The different $K^+$ transporters are characterized by the parameters $g_{KC}$, $g_{HKs}$ and $g_{HKa}$, which indicate their activity relative to the maximal activity of the $H^+$-ATPase. In steady state, the $K^+$ and $H^+$ fluxes through the transporters are represented by (Dreyer et al., 2024):

$$J_{K,KC}^{ss} = \frac{I_{Hmax}}{e_0} \cdot g_{KC} \times \left( V_{ss} - E_K^{ss} \right) \tag{1}$$

$$J_{K,HKs}^{ss} = \frac{I_{Hmax}}{e_0} \times g_{HKs} \times \left( 2 \times V_{ss} - E_H^{ss} - E_K^{ss} \right) \tag{2}$$

$$J_{H,HKs}^{ss} = \frac{I_{Hmax}}{e_0} \times g_{HKs} \times \left( 2 \times V_{ss} - E_H^{ss} - E_K^{ss} \right) \tag{3}$$

$$J_{K,HKa}^{ss} = -\frac{I_{Hmax}}{e_0} \times g_{HKa} \times \left( E_K^{ss} - E_H^{ss} \right) \tag{4}$$

$$J_{H,HKa}^{ss} = \frac{I_{Hmax}}{e_0} \times g_{HKa} \times \left( E_K^{ss} - E_H^{ss} \right) \tag{5}$$

$$J_{HATPase}^{ss} = \frac{I_{Hmax}}{e_0} \times i_p \left( V_{ss} \right) \tag{6}$$

with

$$i_p \left( V_{ss} \right) = \frac{1 - e^{-\left( \frac{F}{RT} \times (V_{ss} - V_{0,pump}) \right)}}{1 + e^{-\left( \frac{F}{RT} \times V_{ss} + 5.4 \right)} + e^{-\left( 0.1 \times \frac{F}{RT} \times V_{ss} + 2.5 \right)}} \tag{7}$$

$$V_{ss} - E_H^{ss} + i_p \left( V_{ss} \right) \times \frac{g_{KC} + g_{HKs} + g_{HKa}}{4 \times g_{HKs} \times g_{HKa} + g_{KC} \times g_{HKs} + g_{KC} \times g_{HKa}} = 0 \tag{8}$$

$$E_K^{ss} = E_H^{ss} - i_p \left( V_{ss} \right) \times \frac{g_{KC} + 2 \times g_{HKs}}{4 \times g_{HKs} \times g_{HKa} + g_{KC} \times g_{HKs} + g_{KC} \times g_{HKa}} \tag{9}$$

and

$$J_K^{ss} = J_{K,KC}^{ss} + J_{K,HKs}^{ss} + J_{K,HKa}^{ss} = 0 \tag{10}$$

$$J_H^{ss} = J_{H,HKs}^{ss} + J_{H,HKa}^{ss} + J_{HATPase}^{ss} = 0 \tag{11}$$

### 2.1. Perturbation of the system

To get an idea about the dynamics of the homeostat, we perturbed the system starting from the stable steady state and introduced perturbations in the electrochemical gradient, that is, in the membrane voltage ($\Delta V = V - V_{ss}$), the potassium gradient ($\Delta E_K = E_K - E_K^{ss}$) and/or the proton gradient ($\Delta E_H = E_H - E_H^{ss}$). These perturbations change the driving forces of transport through each transporter and therefore affect the fluxes. In case of a linear dependence of the flux $J$ on the driving force $\Delta\mu$, the nonsteady-state fluxes are given by (Supplementary Text S1):

$$J_{K,KC} = J_{K,KC}^{ss} + \frac{I_{Hmax}}{e_0} \times g_{KC} \times \left( \Delta V - \Delta E_K \right) \tag{12}$$

$$J_{K,HKs} = J_{K,HKs}^{ss} + \frac{I_{Hmax}}{e_0} \times g_{HKs} \times \left( 2 \times \Delta V - \Delta E_H - \Delta E_K \right) \tag{13}$$

$$J_{H,HKs} = J_{H,HKs}^{ss} + \frac{I_{Hmax}}{e_0} \times g_{HKs} \times \left( 2 \times \Delta V - \Delta E_H - \Delta E_K \right) \tag{14}$$

$$J_{K,HKa} = J_{K,HKa}^{ss} - \frac{I_{Hmax}}{e_0} \times g_{HKa} \times \left( \Delta E_K - \Delta E_H \right) \tag{15}$$

$$J_{H,HKa} = J_{H,HKa}^{ss} + \frac{I_{Hmax}}{e_0} \times g_{HKa} \times \left( \Delta E_K - \Delta E_H \right) \tag{16}$$

The altered fluxes in turn cause changes in $[K^+]_{out}$, $[K^+]_{in}$, $[H^+]_{out}$, $[H^+]_{in}$, and the membrane voltage $V$. The changes in $K^+$ concentrations are governed by the $K^+$ flux:

$$\frac{d[K^+]_{in}}{dt} = \frac{-1}{Vol_{in} \cdot N_A} \times \left( J_{K,KC} + J_{K,HKs} + J_{K,HKa} \right) \tag{17}$$

$$\frac{d[K^+]_{out}}{dt} = \frac{1}{Vol_{out} \cdot N_A} \times \left( J_{K,KC} + J_{K,HKs} + J_{K,HKa} \right) \tag{18}$$

Here, $N_A$ is the Avogadro constant, and $Vol_{in}$ and $Vol_{out}$ are the volumes of the compartments on the respective side of the membrane. Changes in proton concentrations are determined by the $H^+$ flux and the buffer reactions:

$$\frac{d[H^+]_{in}}{dt} = \frac{-1}{Vol_{in} \cdot N_A} \times \left[ J_{HATPase} + J_{H,HKs} + J_{H,HKa} - k_{v,in} \right.$$
$$\left. \times \left( [H^+]_{in} \times [Buf^-]_{in} - K_{s,in} \times [HBuf]_{in} \right) \right] \tag{19}$$

$$\frac{d[H^+]_{out}}{dt} = \frac{1}{Vol_{out} \cdot N_A} \times \left[ J_{HATPase} + J_{H,HKs} + J_{H,HKa} - k_{v,out} \right.$$
$$\left. \times \left( [H^+]_{out} \times [Buf^-]_{out} - K_{s,out} \times [HBuf]_{out} \right) \right] \tag{20}$$

$K_{s,in} = 10^{-pK_{s,in}}$ and $K_{s,out} = 10^{-pK_{s,out}}$ indicate the dissociation constants of the buffer reactions that buffer $pH_{in}$ and $pH_{out}$ to $pK_{s,in}$ and $pK_{s,out}$, respectively. The buffer concentrations, in turn, change

according to:

$$\frac{d[Buf^-]_{in}}{dt} = -\frac{d[HBuf]_{in}}{dt} = \frac{-1}{Vol_{in} \cdot N_A}$$
$$\times k_{v,in} \times \left([H^+]_{in} \times [Buf^-]_{in} - K_{s,in} \times [HBuf]_{in}\right) \tag{21}$$

$$\frac{d[Buf^-]_{out}}{dt} = -\frac{d[HBuf]_{out}}{dt} = \frac{-1}{Vol_{out} \times N_A} \times k_{v,out}$$
$$\times \left([H^+]_{out} \times [Buf^-]_{out} - K_{s,out} \times [HBuf]_{out}\right) \tag{22}$$

Charge transport from one side to the other is a net current

$$I = e_0 \times (J_H + J_K)$$
$$= e_0 \times (J_{HATPase} + J_{K,KC} + J_{K,HKs} + J_{H,HKs} + J_{K,HKa} + J_{H,HKa}) \tag{23}$$

that changes the membrane voltage according to:

$$\frac{dV}{dt} = \frac{-I}{C} = \frac{-e_0 \times (J_H + J_K)}{C} \tag{24}$$

with the membrane capacitance $C$ (unit F).

## 2.2. Kinetic changes

Changes in the parameters $g_X$ are usually not instantaneous and follow chemical reaction kinetics (Dreyer, 2017; Dreyer et al., 2004). To illustrate the effects, we modeled the time course of the change from $g_X^{start}$ to $g_X^{end}$ in the simplest way with mono-exponential relaxation kinetics:

$$g_X(t) = g_X^{end} + \left(g_X^{start} - g_X^{end}\right) \times e^{-t/\tau} \tag{25}$$

with the characteristic time constant $\tau$ that represents the particular features of the transporter, e.g. the relaxation of stretch-activated channels upon a change in the transmembrane pressure difference (Maksaev & Haswell, 2012).

The parameter $g_X$ often depends on other variable parameters, like the membrane voltage $V$, for example. In such a case, a change in $V$ not only leads to the resulting adjustments of the homeostat but also to an additional change in $g_X$ with the subsequent consequences. Nevertheless, the change in $g_X$ and its effects are usually at least one order of magnitude slower (100 ms to 1 s) than the effects of changes in $V$ (10 to 100 ms). To better understand the different contributions of the various parameters, we simulated the changes in the individual parameters in the model scenarios separately.

## 2.3. Computational simulation of the transporter system

To illustrate the dynamic behavior of the K homeostat using concrete examples, the system of differential equations was mathematically simulated using VCell Modeling and Analysis Software developed by the National Resource for Cell Analysis and Modeling, University of Connecticut Health Center (Schaff et al., 1997). The different parameters are detailed in the text and the figures.

## 3. Results

The transporter network of the K homeostat establishes stable steady-state conditions for the membrane voltage $V$ and the $K^+$ gradient $E_K$. Their values are determined by the activities of the different $K^+$ transporters ($g_{KC}$, $g_{HKs}$ and $g_{HKa}$) relative to the activity of the $H^+$-ATPase [equations (8) and (9); (Dreyer et al., 2024)].

Any change in the $g_X$ values, an external voltage stimulus, $\Delta V$, or a change in the transmembrane $K^+$ ($\Delta E_K$) or $H^+$ gradient ($\Delta E_H$) is a disturbance of the steady state condition and causes a reaction of the transporter system. In the following, we will systematically analyze these disturbances and their consequences. It should be noted that changes in one parameter often result in changes in other parameters. For example, a change in membrane voltage also influences the activity of voltage-dependent transporters (Dreyer et al., 2021) or a change in the $K^+$ concentration affects the activity of concentration-dependent transporters (Maierhofer et al., 2024). Such connections and feedback loops make the interpretation of biological systems considerably more difficult, as it is no longer possible to clearly distinguish between cause and effect. The model scenarios presented in our theoretical analyses ultimately aim to convey a deeper understanding of the individual relationships between individual parameters and homeostat behavior. Against this background, it is sometimes a didactic necessity to initially disregard possible feedback effects.

### 3.1. Voltage changes

At first, we considered in a concrete scenario an electrical signal that significantly changed the membrane voltage (by $\Delta V(0) = V(0) - V_{ss}$) but not the $K^+$ gradient ($\Delta E_K = 0$). Such voltage changes have their physiological equivalent in action potentials (Brownlee, 2022; Cuin et al., 2018; Hedrich et al., 2016; Scherzer et al., 2022) but are also induced in optogenetic experiments (Huang et al., 2021; Papanatsiou et al., 2019; Reyer et al., 2020). For illustration, we chose a physiological model scenario, in which a steady state has been established at $V_{ss} = -150\ mV$, $E_K^{ss} = -115\ mV$ ($[K^+]_{in} = 150$ mM, $[K^+]_{out} = 1$ mM), and $E_H^{ss} = 92.1\ mV$ (pH$_{in}$ = 7.2, pH$_{out}$ = 5.6). This could be achieved with the parameter set $g_{KC} = 8$ V$^{-1}$, $g_{HKs}$ = 0.02 V$^{-1}$, and $g_{HKa} = 1.378$ V$^{-1}$. Together with an $I_{Hmax} = 50$ pA and a membrane surface of 1000 μm$^2$ ($C = 10$ pF), the respective $H^+$ and $K^+$ current densities were similar to those observed in guard cells (Dietrich et al., 1998; Lohse & Hedrich, 1992). This system was allowed to equilibrate in its steady state and then challenged with a very short (1 ms) voltage stimulus in the range of $\Delta V = \pm 50\ mV$ (Figure 1a,e). Such a change in the membrane voltage altered the driving forces for $K^+$ and $H^+$, which caused a net current (Figure 1b,f), a net $K^+$ flux (Figure 1c,g) and a net $H^+$ flux (Figure 1d,h; Supplementary Text S2) via the different transporters. A hyperpolarization ($\Delta V < 0$; Figure 1a–d) provoked a negative net current and a $K^+$ and $H^+$ influx, while a depolarization ($\Delta V > 0$; Figure 1e–h) provoked a positive current and effluxes. In this scenario, we have excluded any dependence of the transporter activities on the membrane voltage to clearly work out the effect of the voltage change on the driving forces. The effects of changes in transporter activity that would superimpose the consequences presented here are discussed below.

The net charge transport had a rapid effect on the membrane voltage, with a tendency to restore the original voltage within tens of milliseconds, that is, $\Delta V(t)$ and the ion fluxes varied with time. The time until the steady state ($dV/dt = 0$) was reached, varied with the K-channel activity. With higher activity, the reset time was shorter than with low activity. For comparison, we chose a second scenario, in which the K-channel activity was 10-fold lower than in the first scenario but with the same steady state condition ($V_{ss} = -150\ mV$, $E_K^{ss} = -115\ mV$). This was achieved with $g_{KC} = 0.8$ V$^{-1}$, $g_{HKs}$ = 0.474 V$^{-1}$, and $g_{HKa} = 0.77$ V$^{-1}$ (Supplementary Figure S2). The slightly longer relaxation time in the second scenario was mainly due to the lower net current, which restored the membrane voltage.

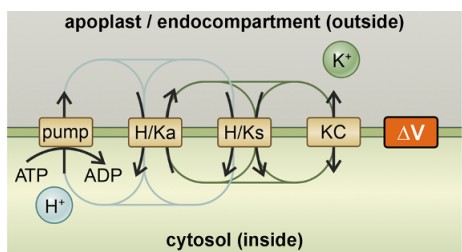

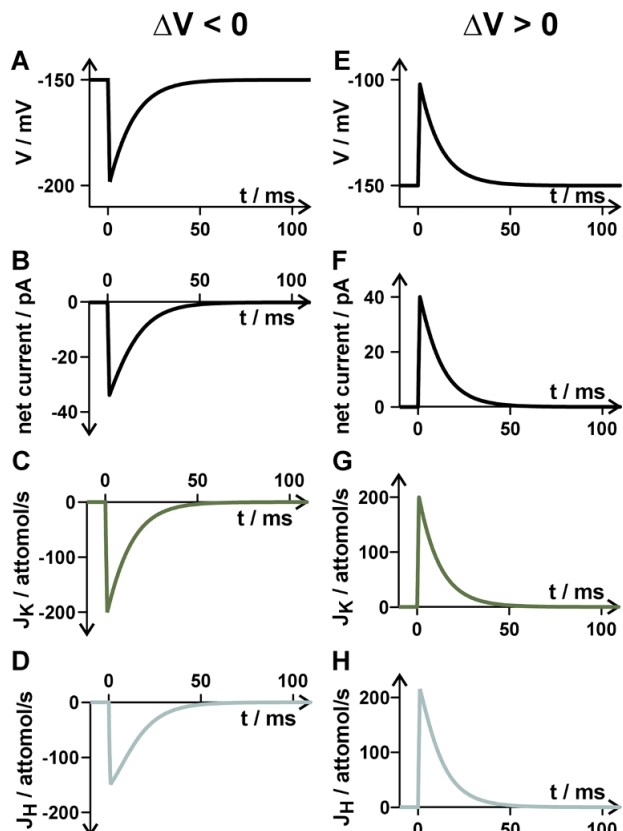

**Figure 1.** Response of the K homeostat to a very rapid voltage pulse (ΔV). The K homeostat in steady state is challenged at *t* = 0 s by a rapid (1 ms) hyperpolarization (a–d) or depolarization (e–h) and is then left to its own devices again. The induced net currents (b, f) cause the membrane voltage to reset over time (a, e). The net currents are carried by net fluxes of $K^+$ ($J_K$; c, g) and $H^+$ ($J_H$; d, h) across the membrane.

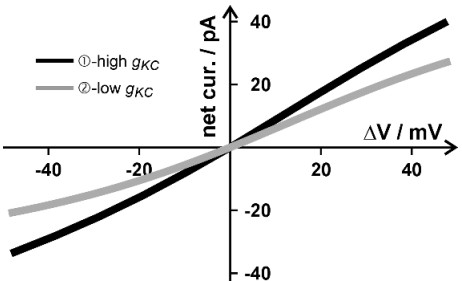

**Figure 2.** Fundamental response of a homeostat to a voltage stimulus. A voltage stimulus (ΔV) induces a net current, which in turn causes a negative restoring force for the membrane voltage. The system is stable at ΔV = 0.

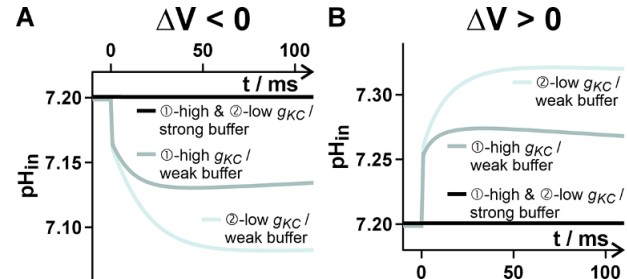

**Figure 3.** Buffer-dependent changes of the internal proton concentration in response to a voltage stimulus. Hyperpolarization (a) or depolarization (b) provoke $H^+$ fluxes that may significantly modify the internal pH (pH$_{in}$). The magnitude of this pH change depends on the buffer capacity of the cell and the composition of the homeostat. Compared are cells with a strong (≥10 mM; black curves) and a very weak (0.1 mM) buffer capacity, as well as a K homeostat with high (①) and one with low (②) K-channel activity.

As expected for a system with fewer K channels, the $K^+$ flux ($J_K$; Supplementary Figure S2c and g) in particular was reduced and was only partially compensated by a larger $H^+$ flux ($J_H$; Supplementary Figure S2d and h).

Irrespective of the scenario, the K homeostat showed a very fundamental short-term behavior of homeostats in general: a voltage change $\Delta V$ induced a net current that was proportional to the voltage change (Figure 2) and which counteracted the deflection from a steady state. The positive current induced by a depolarization ($\Delta V > 0$) drove the membrane voltage in the negative direction while the negative current induced by a hyperpolarization ($\Delta V < 0$) caused a positive shift of the voltage. Thus, the dynamics of the homeostat stabilized the membrane voltage at the steady state value. The more K channels were active in the K homeostat, the stronger was this autoregulative feedback. Current–voltage (I/V) curves of the type shown in Figure 2 are not limited to K homeostats, but a characteristic feature of all types of homeostats. Actually, they

are the fundamental basis for setting the membrane voltage in the steady state.

The $K^+$ and $H^+$ fluxes induced by the voltage stimulus were in the range of hundreds of attomols ($10^{-18}$ mols) per second. Over the time course of the relaxation process, they summed up to a transmembrane net transfer of up to 1–4 attomols (Supplementary Figure S3). To estimate the potential consequences of these at first sight extremely tiny fluxes, it needed to be put in relation to the cellular volume. Considering in our model scenarios a spherical cell with C = 10 pF (cell surface = 1000 μm$^2$), the internal volume would be ~3 pL (3000 μm$^3$). Considering further that the vacuole could occupy about 90% of the internal cellular space, we calculated with a cytosolic volume of 0.3 – 3 pL. A mass transfer of 3 attomols into or out of such a volume would change the concentration by 1 μM (3 pL) or 10 μM (0.3 pL). For $K^+$ with a concentration of $[K^+]_{in}$ = 100 mM, such a change was insignificant. In contrast, for $H^+$ with a resting concentration of $[H^+]_{in}$ = 0.063 μM (pH 7.2) such a change would have implied an alteration of 1500%–15,000% in the absence of any pH buffer system. So far, the cellular simulations were carried out with a potent pH buffer ($[Buf] = [HBuf] \geq 10$ mM). Under this condition, the $H^+$ fluxes did not change pH$_{in}$ in the considered short time interval. However, when we repeated the simulations with 100-fold smaller buffer concentrations ($[Buf] = [HBuf]$ ~0.1 mM), the cytosolic pH was strongly affected (Figure 3). Although this low buffer capacity might not be physiological (Felle, 2001), this example is intended to illustrate the consequences of small buffering capacities. A depolarization alkalinized the cytosol, while a hyperpolarization caused an acidification. The magnitude of the pH changes was different in the two homeostat scenarios. It was

more pronounced in scenario 2 with fewer K-channel activity than in scenario 1 because the smaller $K^+$ fluxes were partially compensated by larger $H^+$ fluxes in the second scenario. Thus, a homeostat has a resetting/stabilizing effect on a disturbance of the membrane voltage. In addition, it could in principle convert a voltage signal into a $pH_{cyt}$ signal. How sensitive this conversion is, depends on the buffer capacity and on the composition of the homeostat. Interestingly, also the feature of converting an electrical into a chemical signal could be generalized to other homeostats. For example, a homeostat with $Ca^{2+}$-permeable transporters could convert a voltage signal into a cytosolic $Ca^{2+}$ signal. Resting $[Ca^{2+}]_{cyt}$ – like $[H^+]_{cyt}$ – is very low and can therefore be quickly changed by fluxes in the range of tens to hundreds of attomoles per second. The time course of the conversion of the electrical signal into a chemical signal depends strongly on the respective chemical buffer capacity of the cell. The lower the buffer capacity, the more sensitive is the conversion process.

### 3.2. Change in $K^+$ concentration

Next, we considered in our model scenarios a sudden change in the concentrations of the permeating ions, that is, in the case of the K homeostat $K^+$ (this section) and $H^+$ (next section). Also these changes affected the driving forces of transport. Although a slower change is more in line with physiological reality, it is by analyzing abrupt changes that one can learn the most about the dynamics of the system. First of all, we changed $[K^+]_{out}$, which affected the $K^+$ gradient ($[K^+]_{out}$ increase $\rightarrow \Delta E_K > 0$; $[K^+]_{out}$ decrease $\rightarrow \Delta E_K < 0$). When we started with the K homeostat in steady state conditions and then doubled or halved $[K^+]_{out}$ we induced a $\Delta E_K = \pm 17.3 mV$. The cellular system responded to such a disturbance in two temporally separated phases (Supplementary Text S3). In a first phase of a few tens of milliseconds, the membrane voltage quickly reached a new, transient peak (i.e. $dV/dt \approx 0$; Figure 4a), while in the second phase, a stable, long-lasting $H^+/K^+$ antiport occurred (Figure 4c,f), which could theoretically last for several tens of minutes (up to hours) with a correspondingly strong pH buffer. To illustrate this process free from secondary pH-effects, we clamped in our simulations $[H^+]_{in}$ and $[H^+]_{out}$. The effects of a limited buffer capacity are illustrated further below. The condition $dV/dt = 0$ implied $J_K = -J_H$, i.e. an electroneutral $H^+/K^+$ antiport [equation (24)]. Thus, in the second phase, after equalization of the membrane voltage, the K homeostat mediated an exchange of $K^+$ for $H^+$, which slowly changed the potassium and proton concentrations (in our simulations clamped) until also here a new steady was established. Exactly, such a $H^+/K^+$ antiport has been observed experimentally in guard cells and mesophyll cells (Li et al., 2024), which underpins the correctness of the theoretical considerations. It should be noted that the $H^+/K^+$ antiport occurred regardless of whether the K homeostat contained antiporters or not. Further inspection revealed that the $H^+/K^+$ antiport was not exactly electroneutral, which is illustrated for a positive $\Delta E_K$ in the following. After the first depolarization caused by a decaying negative net current with an amplitude in the pA-range (Figure 4b), in the following, the net current was not exactly zero but slightly positive with an amplitude in the sub-fA-range (Figure 4e, inset), that is, the $H^+$-efflux was slightly larger in amplitude than the $K^+$-influx. This positive net current led to a very slow return of the membrane voltage to the stable steady state at $V_{ss} = -150 mV$ (Figure 4d). The negative $K^+$ flux ($J_K$, Figure 4c,f) caused an increase of $[K^+]_{in}$ with the tendency of $\Delta E_K \rightarrow 0$ and restauration of the original $E_K^{ss}$, that is, the larger

$[K^+]_{out}$ was compensated by a larger $[K^+]_{in}$. When we repeated the simulations with the K homeostat of the second scenario (fewer K channels), we observed essentially a similar behavior (Supplementary Figure S4a–f). Just the amplitudes of the transient voltage change, the net currents and the $H^+$ and $K^+$-fluxes were smaller indicating that in this case the K homeostat of scenario 1 was more sensitive than that of scenario 2. Qualitatively identical results, but reversed in sign, were obtained when a $\Delta E_K = -17.3 mV$ (reduction of $[K^+]_{out}$) was applied (Supplementary Figure S4g–l).

To illustrate the effect of the pH buffer, we repeated the experiment shown in Figure 4 (clamped $pH_{in}$, i.e., infinite buffer capacity), where $[K^+]_{out}$ has been doubled at $t = 0$, with finite buffer capacities (100 mM, 50 mM, and 10 mM; Kurkdjian & Guern, 1989) (Supplementary Figure S5). When the buffer capacity had no limit (Figure 4), the $H^+/K^+$ exchange proceeded until also $[K^+]_{in}$ had doubled, the $K^+$ gradient was as before ($\Delta E_K = 0$), and the original membrane voltage was restored ($\Delta V = 0$, *i.e.*, V = −150 mV; Supplementary Figure S5, black curves). However, when the buffer capacity reached its limits, the efflux of protons could only be partially compensated and the internal pH value increased (Supplementary Figure S5e; gray curves). This in turn changed the transmembrane proton gradient ($E_H$) and caused the $H^+/K^+$ exchange to terminate earlier (Supplementary Figure S5b and d), so that neither the $K^+$ gradient (Supplementary Figure S5c) nor the membrane voltage (Supplementary Figure S5a) was fully restored. Instead, a new $pH_{in}$-dependent steady-state condition was established. Thus, a homeostat can convert a concentration gradient signal into a voltage and a $pH_{in}$ signal. The sensitivity of this conversion depends on the pH buffer capacity and the composition of the homeostat.

### 3.3. Change in $H^+$ concentrations

Our considerations have shown so far that a K homeostat could process (i) a voltage signal into a $pH_{in}$ signal and (ii) a change in the $K^+$ gradient into a voltage and a $pH_{in}$ signal. As next we considered a change in the proton gradient ($\Delta E_H$). We could have simulated similar scenarios as shown before for $\Delta V$ (Figures 1–3 and Supplementary Figures S2 and S3) or $\Delta E_K$ (Figure 4 and Supplementary Figures S4 and S5). However, closer mathematical inspection revealed that the scenario of $\Delta E_H$ could be explained by a combination of the previously considered perturbations ($\Delta V$ and $\Delta E_K$). Because the steady state of the K homeostat ($V_{ss}, E_K^{ss}$) depends on $E_H$ [equations (8, 9)], a change in this value sets a new steady-state condition of the homeostat ($V_{ss,new}, E_K^{ss,new}$). Consequently, the homeostat, which was originally in dynamic equilibrium ($V_{ss,old}$, $E_K^{ss,old}$), was set by $\Delta E_H$ at $t = 0$ to a state corresponding to a deflection from its new dynamic equilibrium. After the change, the situation of the homeostat could therefore also be explained by $\Delta V = V_{ss,old} - V_{ss,new}$ and $\Delta E_K = E_K^{ss,old} - E_K^{ss,new}$, causing responses such as those described in the previous two sections. In a rapid reaction, the membrane voltage and, under very weak buffer conditions, also $pH_{in}$ were adjusted within tens of milliseconds to a temporary steady state. In the following, the net $K^+$ and $H^+$ fluxes as well as the tiny, but non-zero net current caused by the concentration disequilibrium changed then slowly $[K^+]_{in}$, $[H^+]_{in}$ (depending on the buffer capacity) and $V$ until a new steady state was reached (compare Figures 1–4 and Supplementary Figures S2–S5). Thus, from a mechanistic point of view, the cases of $\Delta E_H$ and $\Delta E_K$ did not differ.

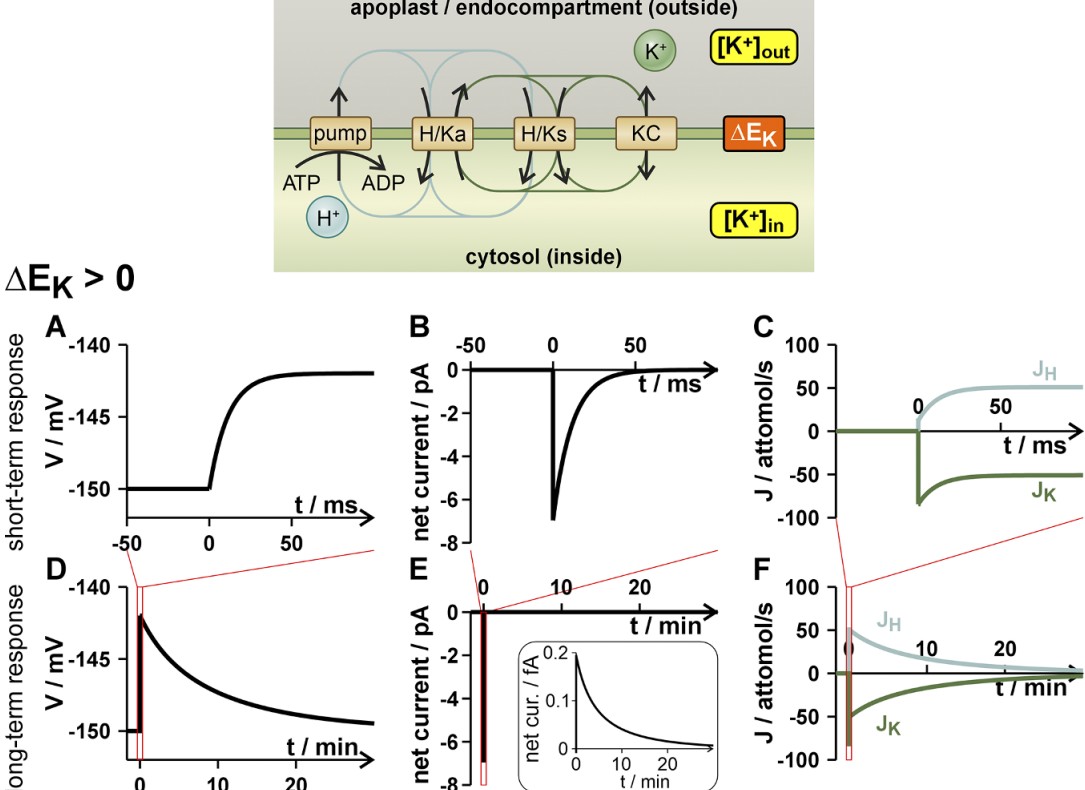

**Figure 4.** Response of the K homeostat to a rapid change in the $K^+$ gradient ($\Delta E_K$). The K homeostat in steady state is challenged at $t = 0$ s by a doubling of $[K^+]_{out}$ ($\Delta E_K > 0$). The induced net currents (b, e) cause the membrane voltage to reset over time (a, d). The net currents are carried by net fluxes of $K^+$ ($J_K$, green) and $H^+$ ($J_H$, blue; c, f) across the membrane. Please note the different answers in the millisecond (short-term response, a–c) and minutes time scale (long-term response, d–f).

### 3.4. Changes in transporter activities (parameters $g_X$)

Finally, we considered what happened if internal or external stimuli modified the activities of the membrane transporters of the homeostat. Physiologically, this happens very frequently, for example, when expression levels change or when posttranslational modifications occur. However, physical parameters such as voltage or hydraulic pressure can also be regulators of the transporter proteins.

As in the previous case, we were able to gain important insights by taking a closer look at the mathematical representation of the homeostat. When being in a steady state, the values ($V_{ss}, E_K^{ss}$) were determined according to equations (8, 9) by the activity of the proton ATPase, the $K^+$ channel, the $H^+/K^+$ symporter and the $H^+/K^+$ antiporter, represented by ($g_{KC}, g_{HKs}, g_{HKa}$). A change in any of the $g_X$ implied a new steady state condition, which was determined by the altered parameter set. The current condition ($V_{ss,old}, E_K^{ss,old}$) could thus be interpreted, as in the case of $\Delta E_H$, as a deviation by $\Delta V$ and $\Delta E_K$ from this new steady state ($V_{ss,new}, E_K^{ss,new}$, Supplementary Text S4).

Changes in $g_X$ might have been caused by an altered expression level, protein turnover, or posttranslational modifications, for instance. These processes are usually rather slow (minutes to hours) suggesting that in these cases also ($V_{ss,new}, E_K^{ss,new}$) are slowly adjusted. But they could also be faster as in the case of wounding when a hydraulic wave is assumed to propagate quickly within the tissues and modifies the activity of tension-regulated channels/transporters (Moe-Lange et al., 2021). To classify the consequences of such quicker processes, we exemplarily altered the parameter of the K channel, $g_{KC}$, in scenarios 1 and 2 consid-

ered before. In scenario 1, we decreased $g_{KC}$ by a factor of 10 to the value of $g_{KC}$ of scenario 2, and in scenario 2, we increased $g_{KC}$ by a factor of 10 to the $g_{KC}$ value of scenario 1. These alterations changed the steady-state values from ($V_{ss,old} = -150\ mV$, $E_K^{ss,old} = -115\ mV$) to ($V_{ss,new1} = -168.3\ mV$, $E_K^{ss,new1} = -7.3\ mV$) and ($V_{ss,new2} = -149.2\ mV$, $E_K^{ss,new2} = -141.5\ mV$), respectively. Instead of an abrupt change of $g_{KC}$, we chose an exponential decay kinetics characterized by a time constant $\tau$. We did this because ensembles of ion channels or transporters, which individually are stochastically acting objects, adapt their activity to external stimuli with exponential kinetics (Dreyer, 2017; Dreyer et al., 2004).

To understand the short-term effects of the $g_{KC}$ changes on the membrane voltage, we first considered the consequences of the equivalent changes in $\Delta V$ and $\Delta E_K$ (Supplementary Text S3). A decrease of $g_{KC}$ caused a $V_{ss,new1} = -168.3\ mV$ more negative than the original $V_{ss,old} = -150\ mV$, that is, $\Delta V = V_{ss,old} - V_{ss,new1} > 0$, and a $E_K^{ss,new1} = -7.3\ mV$ more positive than the original $E_K^{ss,old} = -115\ mV$, that is, $\Delta E_K = E_K^{ss,old} - E_K^{ss,new1} < 0$. Both, $\Delta V > 0$ and $\Delta E_K < 0$ provoked a positive net current (Figures 1f,2 and Supplementary Figure S4h and k) that hyperpolarized the membrane toward a temporary minimum. The time course of this hyperpolarization depended on $\tau$, the reaction time constant of the channel ensemble (Figure 5a). With $\tau = 100$ ms, the first rapid adjustment of the voltage took less than a second, while with a nonunrealistic $\tau = 1$ s (Brüggemann et al., 1999; Dietrich et al., 1998; Maksaev & Haswell, 2012) it could take around 10 s. Please note that the change in $g_{KC}$ provoked a $\Delta E_K < 0$, which caused a temporary membrane voltage peak (−182 mV, Figure 5a) that was negative of the new final steady state ($V_{ss,new1} = -168.3\ mV$), a phenomenon observed

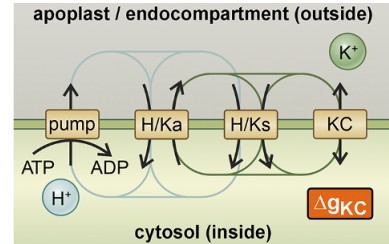

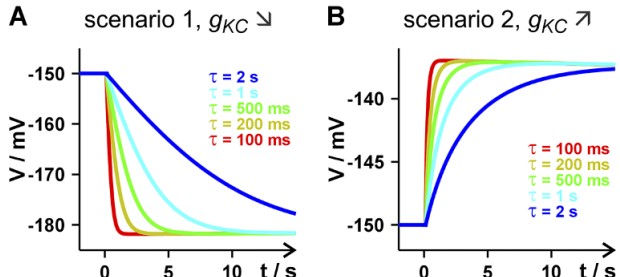

**Figure 5.** Changes in the membrane voltage as a response to changes in $g_{KC}$ of the K homeostat. Shown is the first phase of the biphasic adjustment of the voltage. (a) In scenario 1, $g_{KC}$ started at $t = 0$ to decay exponentially with time constant $\tau$ toward the $g_{KC}$ value of scenario 2 (decay by factor 10). (b) In scenario 2, $g_{KC}$ started at $t = 0$ to increase with exponential saturation with time constant $\tau$ toward the $g_{KC}$ value of scenario 1 (increase by factor 10).

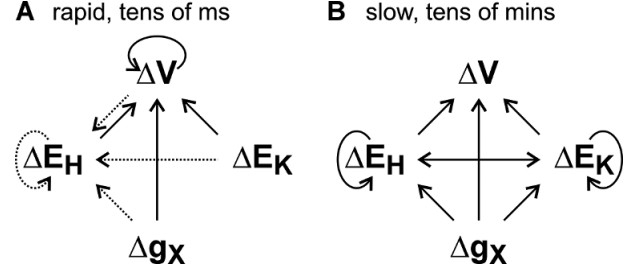

**Figure 6.** Schematic overview of the dynamic properties of the K homeostat. Changes in the membrane voltage ($\Delta V$), the potassium gradient ($\Delta E_K$), the proton gradient ($\Delta E_H$), and the transporter activities ($\Delta g_X$) influence each other. The effects manifest rapidly within tens of milliseconds (a) and slowly on a time scale of tens of minutes (b). The interactions shown with dashed arrows in (a) only occur if the buffer capacity is absent or low. In addition to the shown interactions, there might also be feedbacks on an intermediate time scale from $\Delta V$, $\Delta E_K$, and/or $\Delta E_H$ on $\Delta g_X$ that are not indicated; for example, in the case of voltage-gated ion channels, where $\Delta V$ feeds back on $\Delta g_X$ (Dreyer et al., 2004).

already before (Supplementary Figure S4g and j). This imbalance would equilibrate in the slow phase of the adjustment (tens of minutes) that is dominated by a change of the concentrations due to the net fluxes of $K^+$ and $H^+$ (compare Supplementary Figure S4g–l).

Similar conclusions as those illustrated for a decrease of $g_{KC}$ could also be obtained when increasing $g_{KC}$, albeit with inverse signs. Compared to the starting conditions ($V_{ss,old} = -150\ mV$, $E_K^{ss,old} = -115\ mV$) the new steady state $V_{ss,new2}$ was more positive i.e. $\Delta V = V_{ss,old} - V_{ss,new2} < 0$, and $E_K^{ss,new2}$ more negative than the starting value $E_K^{ss,old}$, i.e. $\Delta E_K = E_K^{ss,old} - E_K^{ss,new2} > 0$. Both deviations from the steady state caused a negative current (Figures 1b and 2, S4b and e) that triggered a temporary depolarization of the membrane voltage (Figure 5b). In the long term (tens of minutes), the membrane voltage would repolarize to the new steady state ($V_{ss,new2} = -149.2\ mV$) alongside with an increase of $[K^+]_{in}$ ($E_K \rightarrow E_K^{ss,new2}$) due to the net $K^+$ influx (Supplementary Figure S4c and f). The long-term fluxes were strongly influenced by the buffer capacity of the cell (compare Supplementary Figure S5), as a $K^+$ influx was accompanied by an $H^+$ efflux and a $K^+$ efflux by an $H^+$ influx. In summary, these considerations indicated that a change in the activity of an ion channel type involved in a homeostat is a strong signal that affects the membrane voltage, the pH conditions and in the long-term the ionic conditions of the cell.

## 4. Discussion and conclusion

Homeostats are networks of differentially energized transporters that transfer the same sort of ions/molecules across a membrane. A previous study argued on proper thermodynamic grounds that homeostats are the hidden rulers of homeostasis in plant cells (Dreyer et al., 2024). We have now analyzed their particular dynamic properties and propose that homeostats also play a key role in electrical, hydraulic and chemical signaling, such as pH and calcium signaling.

Exemplarily for the K homeostat we could show that homeostats equip the membrane with a self-regulatory, voltage-stabilizing property (Figure 2), in which a depolarization causes a positive net current that drives the membrane voltage in the negative direction, while a hyperpolarization provokes the opposite. Such a feature is not limited to K homeostats. In fact, it can be generalized. Such a characteristic curve has already been determined experimentally for a $Ca^{2+}$ homeostat in vacuoles (Dindas et al., 2021), indicating that also there several differentially energized transporter types involved in $Ca^{2+}$ transport form a new functional unit. Also other types of homeostats, for example, anion homeostats (Dreyer, 2021; Dreyer et al., 2022; Geisler & Dreyer, 2024; Li et al., 2024), would show the same current–voltage characteristic. Related to this general feature is the rapid stabilization of the membrane voltage that takes tens of milliseconds (Figure 1). In the same time frame also abrupt changes in the $K^+$ ($\Delta E_K$) or the $H^+$ gradient ($\Delta E_H$) or changes in transporter activities ($\Delta g_X$) affect the voltage (Figure 6a). While rapid voltage changes ($\Delta V$) are an important feature in electrical signaling, e.g. by action potentials (Brownlee, 2022; Cuin et al., 2018; Hedrich et al., 2016), abrupt changes of the other parameters ($\Delta E_K$, $\Delta E_H$ or $\Delta g_X$) are not necessarily realistic. Usually, these alterations take place more slowly. We nevertheless examined exemplarily instantaneous settings of $\Delta E_K$ (Figure 4 and Supplementary Figures S4 and S5) to illustrate the fast dynamics of the system and show that this case explains in principle also the features of rapid changes of $\Delta E_H$ or $\Delta g_X$ (Supplementary Text S2 and S4). In particular the latter (rapid $\Delta g_X$) may occur during the optogenic control of ion channels artificially introduced in plants (Christie & Zurbriggen, 2021; Ding et al., 2024; Huang et al., 2021; Huang et al., 2023; Konrad et al., 2023; Papanatsiou et al., 2019; Reyer et al., 2020). Our analyses may therefore form a theoretical basis for these experiments.

In addition to the ad hoc changes, we also chose for $\Delta g_X$ a physiological scenario with an exponential time course. Membrane transporters such as ion channels behave stochastically in response to regulatory parameters. This manifests itself in exponential activation and deactivation time courses of the channel ensemble (Dietrich et al., 1998; Dreyer, 2017; Dreyer et al., 2004). In order to exploit the detailed knowledge from the analysis of abrupt changes, these time courses can be considered to be subdivided into small time intervals with stepwise changes of $g_X$. The membrane voltage adjusts within a few milliseconds to the condition determined by

the actual $g_X$ value and thus follows the delayed time course of the $\Delta g_X$ change (Figure 5).

The scenario shown in Figure 5 is not limited to K homeostats and could explain in general terms the principle of basic sensing mechanisms in plants allowing them to react to a diverse set of physical and chemical stimuli, such as a hydraulic stimulus, or hormones and second messengers. The corresponding homeostat only needs to comprise one type of transporter that reacts sensitively to the respective stimulus. The stimulus alters the activity of the transporter and thus deflects the homeostat from its steady state. This in turn causes $\Delta V$ as the first rapid reaction, possibly followed also by pH (and $Ca^{2+}$) signals.

The ion fluxes that modify the membrane voltage in the short term are generally not large enough to significantly change the cellular concentrations. With two exceptions: $H^+$ and $Ca^{2+}$. The cytosolic concentration of these two ion species is in the tens to hundreds of nanomolar range (Brownlee & Wheeler, 2025; Felle, 2001; Kader & Lindberg, 2010; Köster et al., 2022; Kudla et al., 2018; Luan & Wang, 2021) and could therefore fundamentally change by the ion fluxes quantitatively estimated in this study (Supplementary Figure S3). These fluxes might be fully or partially absorbed by the cellular buffer systems (Brownlee & Wheeler, 2025; Feng et al., 2020; Kosegarten et al., 1997; Niñoles et al., 2013; Oja et al., 1999; Pfanz & Heber, 1986; Wegner et al., 2021; Zhou et al., 2021). Nevertheless, the buffer capacities for both ions are likely to be different, with the capacity for $H^+$ presumably being higher than for $Ca^{2+}$. Thus, we hypothesize that in homeostats involving calcium the $Ca^{2+}$-response accompanied to a rapid voltage change is faster than the suppressed or delayed $H^+$-response shown in this study (Figure 3). Although we have not analyzed $Ca^{2+}$ homeostats here (but initial analyses were presented in Dreyer et al., 2022), we can still state that fast signals ($\Delta V$) are differentially converted to $Ca^{2+}$ and $H^+$ signals, as the larger pH buffer capacity of the cell can attenuate and delay the $H^+$ signal. Thus, a similar translation mechanism ($\Delta V \rightarrow pH_{cyt}$ / $\Delta V \rightarrow Ca^{2+}_{cyt}$) can nevertheless give rise to a multitude of finely tuned cellular signals.

In addition to the short-term responses, there is also a mutual long-term interplay between $\Delta V$, $\Delta E_K$, $\Delta E_H$ and $\Delta g_X$ that take place within tens of minutes (Figure 6b). These changes are mainly due to the slow change of concentrations caused by the net $K^+$ and $H^+$ fluxes (Figures 4f and Supplementary Figure S4f, 1 and S5). Interestingly, in these long-term fluxes the K homeostat mediates an almost electroneutral $K^+/H^+$ exchange even in the absence of functional $H^+/K^+$ antiporters. This feature of K homeostats has been observed experimentally already (Li et al., 2024). Thus, long-term uptake of potassium is always accompanied by an efflux of protons, which would quickly overload the chemical pH buffers of the cells if it is not compensated for in some other way. Here, anion (A) homeostats step in, which mediate an electroneutral $A^-/H^+$ symport (Li et al., 2024). Together, K and A homeostats form a flexible functional super-unit that not only manages the uptake of these ions but also participates in the pH-buffering process (see Graphical Abstract). Thus, the common view that $Cl^-$ (or another anion like $NO_3^-$) is the counterion of $K^+$ guaranteeing electroneutrality is just half of the truth and does not acknowledge the flexibility in pH control by the two involved homeostats.

As indicated, the coupling of different homeostats represents a further level of dynamic complexity. Changes in the setting of one homeostat, that are limited to this homeostat at first glance (e.g. $\Delta E_K$ for the K homeostat) also affect all other connected homeostats, with far-reaching physiological implications. As an example, recently a coupling between the auxin and anion

homeostats has been proposed as the reason why plants use two different auxin exporters, one of which even requires the energy from ATP hydrolysis for a downstream transport (Geisler & Dreyer, 2024). The different homeostats "communicate" with each other predominantly via the membrane voltage, the cytosolic pH, but also by $Ca^{2+}$-regulated protein kinases (Dong et al., 2021; Hamel et al., 2014; Kudla et al., 2018; Wang et al., 2023). We are only just beginning to understand the grammar of this sophisticated language. The long-term simulations presented in this study should therefore not be overinterpreted as we have considered a homeostat in isolation. In real life, cellular countermeasures involving other homeostats and regulatory feedbacks will certainly avoid extreme values for $[K^+]_{in}$ and $pH_{in}$. In this context, it should also be noted that non or poorly selective transporters can link different types of homeostats. For example, if we consider the K homeostat from this work and add e.g. $K^+$- (but also $Na^+$-) permeable channels of the HKT type (Riedelsberger et al., 2021) then the K homeostat is linked to the Na homeostat (HKT & SOS1/NHX; Gámez-Arjona et al., 2024) and forms another new super-unit. All these new units act according to the basic rules shown in Figures 2 and 6. However, their particular characteristics and the question of how one ion can influence the homeostasis of the other still need to be explored in more detail in the future.

In summary, our analyses strongly suggest that it is not individual transporters but the dynamic properties of homeostats that underlie electrical, chemical, hydraulic, pH and calcium signaling in plants. Homeostats are universal building bricks (like LEGO) that then group into larger units to accomplish the complicated tasks of physiology.

**Supplementary material.** The supplementary material for this article can be found at http://doi.org/10.1017/qpb.2025.6.

**Data availability statement.** The authors confirm that the data supporting the findings of this study are available within the article and its supplementary materials. Raw data sets are available from the corresponding author, ID, upon reasonable request.

**Author contributions.** I.D. conceived the study. All authors contributed to the design of the study. I.D. conducted the data analyses. L.C.-A., T.R.-R., T.R.-G., and M.E.R.-M. participated in data visualization. All authors contributed to writing and editing of the manuscript and approved its final version.

**Funding statement.** This work was supported by the Agencia Nacional de Investigación y Desarrollo de Chile (ANID), grant No. Anillo-ANID ATE220043 (the multidisciplinary center for biotechnology and molecular biology for climate change adaptive in forest resources; CeBioClif) to I.D. and by Fondo Nacional de Desarrollo Científico, Tecnológico y de Innovación Tecnológica (FONDECYT/Chile), grant no. 11251254 to M.E.R.-M., no. 1210920 to E.M, no. 1220504 to I.D., no. 1240167 to J.R.

**Competing interest.** The authors declare none.

**Open peer review.** To view the open peer review materials for this article, please visit http://doi.org/10.1017/qpb.2025.6.

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
