## [Reviewer Report]

A logical continuation of the formulation of the model of a K+ homeostat in their previous studies is the testing of its behavior upon perturbations of membrane voltage, sudden changes of external K+ and H+ concentrations, and the turnover rate of individual transporters (here,as an example, ensemble conduction of a population of K+ channels, which could be, for instance, artificially evoked in an optogenetic experiment). Results and conclusions have a fundamental character and represent an important step in a construction of a “superhomeostat”, which combines diverse homeostat circuits.

I have few queries:

-An important conclusion is a “voltage-stabilizing property” of the K+ homeostat. No doubts, it should be true for the circuit which includes H+ pumps, H+/K+ symporters and antiporters, and K+-selective channels. But, what about nonselective cation channels, which hardly differentiate between K+ and Na+? Or, worse, with a low cation-anion preference? Experimentally, there are proofs for long (tens of minutes-days) lasting membrane depolarizations upon strong salt or oxidative stress. Apparently, K+ homeostat fails in this case. Should it be meaningful to specify these limitations, when a disturbance is beyond a capacity of a K+ homeostat?

- Lines 232-233, this very low level for pH buffer capacity seems unrealistic.

-Line 433. I would disagree. Intracellular buffer for Ca2+ (a plethora of Ca2+ binding molecules including proteins/ Ca2+ sensors), of course, exists, although in plants, up to my knowledge, there were no systematic studies to evaluate its capacity. In my opinion, its approximation may be a submillimolar value for the cytoplasm.

- Please, check, is it correct to use the term “equilibrium” in lines 313, 314, 420, when it is apparently speaking about steady state condition.

---

## [Reviewer Report]

This theoretical study by Contador -Álvarez et al. is a follow-up work to a previous publication in the same journal dealing with the K+ homeostat (and elaborating on previous papers on this issue by the corresponding author, Ingo Dreyer). In that previous work the homeostat was introduced as a tool to analyse K+ and H+ concentrations and fluxes under steady state conditions. K+ and H+ homeostasis were shown to result from the interplay of a H+ ATPase, a K+ conductance and H+/K+ symport and antiport, also including H+ buffering. Here, the response to a perturbation of the system is analysed with the same computational tool, analysing the response to sudden changes in (i) membrane voltage, (ii) K+ concentration, and (iii) K+ membrane conductance. This is indeed a logical step forward. The simplicity of this approach proved to be a great advantage for extending our basic knowledge and understanding of membrane transport in plants, but it also comes with a severe limitation: The scenarios are rather artificial, since they refer to a cell with only K+ and H+ as permeable cations, and membrane-impermeable anions required for charge balance, a situation we won’t find in nature. The authors are aware of this limitation, since they state at the end of the discussion that “The long-term simulations presented in this study should therefore not been overinterpreted as we have considered a homeostat in isolation. In real life, cellular countermeasures involving other homeostats and regulatory feedbacks”. However, it is conceivable to design model systems with exactly these properties, e.g. oocytes co-expressing this set of transporters (and maybe inhibiting native anion channels); maybe this should still be pointed out explicitly.

I am sure the authors will explain to us in the near future how K+, H+ and Ca2+ signatures come about, and how they can be decoded by the cell.

I have a number of critical points to raise:

l. 62 …as an example

l. 172 some K+ is transported, so better say: effects on [K+] are negligible..

l. 170 onwards: The homeostat is used in this scenario to analyse the response of the membrane to a stepwise change in voltage at a timescale up to 100 ms. However, this leads us to a fundamental problem: It was not designed for that purpose, but rather, to describe a steady state. Hence, although it does include the kinetics of the buffer reactions adjusting pH, but it doesn’t include the kinetics of the K+ channel and the transporters which will respond with a delay to a stepwise change in voltage (and. likewise, K+). For an ion channel this delay will be in the range of maybe 10 to 100 ms, as numerous patch clamp experiments and analysis in oocytes after heterologous expression have shown, and for a H+ K+ cotransporter such as HAK5 I expect similar kinetics (though I didn’t find any data in the literature). As for a response to a change in K+ concentration, a recent publication by Maierhofer et al. (2024; Nature Communications 15.1; 8558) revealed that it takes even seconds until a new steady state is reached (refers to the next experiment). So the curves shown here are of little practical significance and rather show the properties of the algorithm used here; in the model applied to this scenario all transporters respond instantaneously to a change in parameters. The authors are in principal aware of that problem later saying that “Changes in the parameters gx are usually not instantaneous and follow chemical reaction kinetics” (l. 145), but they treat this issue separately instead of including it when considering changes in voltage and K+ concentration.

Insets to figures: In addition to membrane transport, circular fluxes are shown. The arrows are superimposed but don’t match, which makes the figure look messy and over-complicated.

l. 401: “…are not necessarily realistic…”

l. 411: …manifest themselves in group behaviours, leading to exponential activation and deactivation time courses…

l. 413 ff: This paragraph is a little awkward and needs rephrasing: e.g. we can subdivide the gp timecourse to infinitesimal stepwise changes with delta V following delta gp…

The scenario described here does not only apply to pressure sensing, but to any other (chemical) stimulus, such as hormone, second messengers…

l. 435 There is Ca2+ buffering as well! Bound Ca2+ is exceeding free Ca2+ by orders of magnitude.

l. 450 …long-term interplay between…

l. 455 It seems that H+ can serve as a balance between K+ and anion homeostat?!

Supplement:

Fig S5E: I really have a problem to understand the pH profiles. I would expect an s-shaped curve, with slow pH change at the beginning when the buffer capacity is at maximum and an increase in slope until the buffer is ‘broken’ and a rapid transition to a new steady state when the H+ flux ends.

Text S1, figure S7: mx and sx correspond to chord conductance and slope conductance. The latter has little significance, whereas the former is, from a thermodynamic standpoint, the relevant parameter. Hence, approximating the curve with a Taylor series is not of much practical use either.

---

## [Editor Report]

Dear Ingo et al, 

Thank you for submitting your manuscript on your exciting work on homeostat dynamics to QPB. The reviewers and I are positive about this submission but feel the work could be strengthened by clarifying a number of points (see full reviews for some excellent comments and questions). 

I look forward to receiving your revised manuscript. 

With best wishes

Richard

---

## [Reviewer Report]

The authors have addressed my previous concerns properly and convincingly. I wish to congratulate them on this work!

My only objection refers to the use of the homeostat for modelling short-term changes in voltage and ion concentrations. For modelling K+/H+/Ca2+ signatures, which is the ultimate goal of this work, the authors will have to amend their homeostat such that the complex, short-term interplay of the parameters (voltage, ion concentrations…) is reflected in a still more realistic way.

---

## [Editor Report]

Hi Ingo, 

Please accept my congratulations on this excellent contribution. We are delighted you chose QPB for this work. 

There is a very helpful comment from one of the reviewers regarding the use of your model for short time-scales that you may wish to consider and address before uploading the final version of your manuscript. 

With best wishes

Richard